# Combination of Local Ablative Techniques with Radiotherapy for Primary and Recurrent Lung Cancer: A Systematic Review

**DOI:** 10.3390/cancers15245869

**Published:** 2023-12-16

**Authors:** Paolo Bonome, Donato Pezzulla, Valentina Lancellotta, Anna Rita Scrofani, Gabriella Macchia, Elena Rodolfino, Luca Tagliaferri, György Kovács, Francesco Deodato, Roberto Iezzi

**Affiliations:** 1Radiation Oncology Unit, Responsible Research Hospital, 86100 Campobasso, Italy; donato.pezzulla@responsible.hospital (D.P.); gabriella.macchia@responsible.hospital (G.M.); francesco.deodato@responsible.hospital (F.D.); 2Dipartimento di Diagnostica per Immagini, Radioterapia Oncologica ed Ematologia, UOC Radioterapia Oncologica, Fondazione Policlinico Universitario “A. Gemelli” IRCCS, 00168 Rome, Italy; valentina.lancellotta@policlinicogemelli.it (V.L.); luca.tagliaferri@policlinicogemelli.it (L.T.); 3Dipartimento di Diagnostica per Immagini, Radioterapia Oncologica ed Ematologia, UOC Radiologia d’Urgenza ed Interventistica, Fondazione Policlinico Universitario “A. Gemelli” IRCCS, 00168 Rome, Italy; annarita.scrofani@guest.policlinicogemelli.it (A.R.S.); roberto.iezzi@policlinicogemelli.it (R.I.); 4Dipartimento di Diagnostica per Immagini, Radioterapia Oncologica ed Ematologia, UOC Radiologia Addomino-Pelvica, Fondazione Policlinico Universitario “A. Gemelli” IRCCS, 00168 Rome, Italy; elena.rodolfino@policlinicogemelli.it; 5Gemelli-INTERACTS, Università Cattolica del Sacro Cuore, 20123 Rome, Italy; kovacsluebeck@gmail.com; 6Radiology Institute, Università Cattolica del Sacro Cuore, 00135 Rome, Italy

**Keywords:** lung cancer, SBRT, RT, percutaneous image-guided local tumor ablation (LTA), combining LTA-RT

## Abstract

**Simple Summary:**

The aim of this review was to evaluate the feasibility and the effectiveness of radiation therapy combined with local tumor ablation therapy in the treatment of primary and recurrent lung cancer in terms of toxicity profile and local control rate. Six studies featuring a total of 115 patients and 119 lesions were selected, showing encouraging outcomes that appear to be promising in terms of toxicity profile. Further prospective studies are need to better delineate combining LTA-RT treatment benefits in this setting.

**Abstract:**

In patients with early-stage or recurrent NSCLC who are unable to tolerate surgery, a benefit could derive only from a systemic therapy or another few forms of local therapy. A systematic review was performed to evaluate the feasibility and the effectiveness of radiotherapy combined with local ablative therapies in the treatment of primary and recurrent lung cancer in terms of toxicity profile and local control rate. Six studies featuring a total of 115 patients who met eligibility criteria and 119 lesions were included. Three studies evaluated lung cancer patients with a medically inoperable condition treated with image-guided local ablative therapies followed by radiotherapy: their local control rate (LC) ranged from 75% to 91.7% with only 15 patients (19.4%) reporting local recurrence after combined modality treatment. The other three studies provided a salvage option for patients with locally recurrent NSCLC after RT: the median follow-up period varied from 8.3 to 69.3 months with an LC rate ranging from 50% to 100%. The most common complications were radiation pneumonitis (9.5%) and pneumothorax (29.8%). The proposed intervention appears to be promising in terms of toxicity profile and local control rate. Further prospective studies are need to better delineate combining LTA-RT treatment benefits in this setting.

## 1. Introduction

Lung cancer is one of the most frequent cancers in the world and the main cause of cancer mortality [1]. Traditional treatment for early stage non-small cell lung cancer (NSCLC) is surgical excision with or without thorough lymph node assessment. However, due to severe medical comorbidities, 20% of early-stage NSCLCs have been estimated to be unable to tolerate surgery [2,3]. Only systemic therapy or a few other forms of local therapy could aid these people. As a result, novel local ablative method modalities have emerged to strengthen our therapeutic arsenal [4].

Percutaneous image-guided local tumor ablation (LTA), which includes radiofrequency (RFA), microwave ablation (MWA), and cryoablation (CA), is one of them. LTA, which was first described in clinical trials in 2000 [5], is a minimally invasive approach for the local therapy of lung cancer with encouraging results [6,7]. Another option, as indicated by the National Comprehensive Cancer Network (NCCN) guidelines [8], is stereotactic body radiation (SBRT): a conventional treatment for medically inoperable patients whose efficacy, minimal toxicity, and satisfying local disease control are supported by multiple studies [9,10,11].

Indeed, radiotherapy (RT) and LTA use completely different mechanisms: the former is most effective against well-oxygenated cells in the periphery of the tumor and less effective at eradicating more hypoxic cells in the core, whereas LAT targets the core but is less effective in the periphery due to increasing heat sink effects [12,13,14]. In particular, because of the complimentary activities of these two techniques, some authors hypothesized that combining them in different settings, including pulmonary diseases, could result in a probable synergic result [15,16,17].

Despite these initial intriguing findings, the role and process of integration between RT and LAT are not completely characterized in the existing research, and the available data are inadequate, as they are characterized by a small sample size and heterogenous procedures.

In light of this, we conducted a systematic analysis to assess the feasibility and effectiveness of RT combined with LAT in the treatment of primary and recurrent lung cancer in terms of toxicity profile and local control rate.

## 2. Materials and Methods

### 2.1. Eligibility Criteria

Prospective and retrospective studies were included in this analysis. We used the following inclusion criteria: English language, full-text articles, patients treated with combined LTA-RT, presence of detailed toxicity and local control data. In addition, we used the following exclusion criteria: only abstracts, letters, proceedings from scientific meetings, editorials, expert opinions, reviews without original data, studies lacking toxicity and/or safety outcomes, repetitive data, animal studies, studies with fewer than 5 patients, and studies that included combination different than LAT and SBRT, such as chemotherapy, immunotherapy, or surgery.

### 2.2. Information Sources

This systematic review was performed following recommendations from the Preferred Reporting Items for Systematic Reviews and Meta-Analyses (PRISMA). A comprehensive search was conducted in PUBMED, MEDLINE, SCOPUS and Google Scholar to identify relevant published studies that confirmed the feasibility of integration between January 1999 and December 2022.

### 2.3. Search Strategy

Keywords used were: (NSCLC or lung cancer or lung neoplasm) AND (integration or combination or followed) AND (early stage or primary) AND (recurrent or relapse) AND (radiotherapy or radiation therapy or SBRT or IMRT) AND (LTA or radiofrequency ablation or microwave ablation or cryoablation). The computer search was supplemented manually using reference lists for all available review articles, primary studies, meeting abstracts, and bibliographies of books to identify studies not encountered in the computer search.

### 2.4. Selection Process

Retrieved records underwent title-and-abstract review and then full-text review. Two independent researchers (PB and AS) screened all the studies in duplicate using the eligibility criteria reported above. A third reviewer (DP) rechecked the articles when confronted with discrepancies. Three independent reviewers performed data extraction (PB, AS, DP). Reasons for exclusion at full-text review were recorded. Disagreements among reviewers were infrequent (<20%) and were resolved by discussion.

### 2.5. Data Items

The following data were included: author, year, study design, LTA techniques data (RFA or MWA or Cryoablation), radiation treatment data (i.e., type, fractionation, total dose), clinical/radiological treatment responses, follow-up time, toxicities, local control (LC), defined as response to the treatment until last follow-up or patient’s exitus and overall survival (OS), calculated from the time of treatment until the last follow-up or patient’s exitus survival time at the moment of the treatment.

### 2.6. Quality Assessment

The quality assessment score of included studies was assessed according to a checklist for the quality appraisal of case series studies produced by The Institute of Health Economics (IHE) [18].

### 2.7. Statistical Analysis

Statistical analysis was performed using Review Manager (RevMan) (computer program) Version 5.

Heterogeneity across studies was examined by I^2^ statistic. Studies with I^2^ statistic values of 0–50%, 50–75%, and >75% were considered to have low, moderate, and high heterogeneity, respectively [19]. A forest plot for a post hoc meta-analysis to display the association between lesions size and LC after the combined therapy was generated. We used random-effects models because there was great subjectivity given the lack of related control groups in the non-comparative studies and a tendency toward high heterogeneity.

### 2.8. Review Registration

The review was registered on the Open Science Framework (OSF), obtaining the following registration DOI: https://doi.org/10.17605/OSF.IO/VXGK9 (accessed on 3 November 2023).

## 3. Results

A total of 634 citations were retrieved; 600 of these were removed because they were limited to LTA or RT and focused on integration between the two techniques. The remaining 34 studies were evaluated using their entire texts. Following the rejection of studies with an inappropriate population, therapy, or providing insufficient data (N = 28), six papers were finally selected based on the inclusion criteria outlined above; more information is shown in Figure 1.

Except for Steber et al. [15], a prospective phase 2 study that closed early due to delayed enrollment, all of the studies chosen [16,17,20,21,22] were retrospective. The investigations included 115 patients and 119 lesions in total. The sample size for the majority of these experiences ranged from six patients [22] to 41 patients [16]. Except for Brooks et al. [22], all of the studies included age and gender information. The median age ranged from 55 to 93 years with a 62/47 male/female ratio.

Three studies [15,16,17] evaluated lung cancer patients with a medically inoperable condition treated with LAT followed by radiotherapy, while another three studies [20,21,22] experienced LTA as a salvage option for patients with locally recurrent NSCLC after RT. More details on the patients’ characteristics are reported in Table 1.

### 3.1. First Group: Image-Guided LAT Followed by Radiotherapy

In the first group of studies, 77 patients with early-stage NSCLC were assessed with a male/female ratio of 42/35 and a median age ranging from 55 to 93 years. The patients were staged as follows: stage IA (43 points), stage I B (28 points), stage II B (3 points), and staging data were unavailable for three patients. All patients received image-guided LAT (73 RFA and 4 MWA) before radiotherapy.

RFA was employed by Dupuy et al. [17]: the mean impedance was 72 ohms (range 42 to 11), the mean current was 1.6 amps (range 1.2 to 2.0 amps), and the post-RFA temperatures were greater than 60 °C (range 76.4 °C/62 to 85 °C) with a treatment time of 6.8 min (range 2 to 12). Grieco et al. [16] used RFA with a baseline impedance of 72.7 ohms (range 40–69), power 128.8 W (range 10–196) achieving temperature >70 °C (range 38–94 °C) with a mean treatment time of 6.3 min (range 1–12), whereas MWA had a power (W) of 47.5 (range 45–60) and treatment time of 8.4 min (range 2–10).

Information on the RT technique was available in all three analyzed studies. In the majority of cases (63/77, 81.8%), external beam radiotherapy (EBRT) was performed as three-dimensional conformal radiotherapy (3DCRT) in 51 patients [16,17], hypofractionated radiotherapy (HFRT) without any indication on the type of radiotherapy technique in 9 [15], and stereotactic body radiotherapy (SBRT) in three [15]. In addition, 14 out of 77 patients (18.1%) underwent interventional radiotherapy (IRT, also called brachytherapy). Thirteen patients received high-dose rate IRT with an Iridium ^192^Ir source through an interstitial catheter, and one patient received low-dose rate IRT with 12 permanent iodine ^125^I seeds placed through an interstitial applicator [16]. In all papers, data on total dose and fractionation were described. The most commonly used RT regimen was 66 Gy in 33 fractions (fx) (79.3%) [16,17], which was followed by nine patients receiving 70.2 Gy in 26 fx (14.2%) [15] and one patient receiving 50 Gy in 25 fx [16]. An approach with SBRT was used in three cases with a total dose of 54 Gy in three fractions [15].

Data on tumor size, radiological response evaluation, and median follow-up time are shown in Table 1.

The local control rate (LC) ranged from 75% to 91.7% [15,16,17], with only 15 patients (19.4%) reporting local recurrence after combined modality treatment [15,16,17]. In the study of Grieco et al. [16], local recurrence occurred in 11.8% of lesions smaller than 3 cm after an average of 45.6 ± 4.1 months and in 33.3% of the larger lesions after an average of 34 ± 7.8 months.

Steber et al. [15] reported a median OS value of 53.6 months, while Dupuy et al. [17] reported a mean OS of 26.7 months and rates of cancer-specific survival (CSS) at 12, 24, and 60 months of 83%, 50%, and 39%, respectively. Grieco et al. showed an average OS rate of 34.7 ± 5.4 months if LAT was combined with IRT and of 42 ± 6 months if it was associated with RT [16].

The adverse events and the associated grade of toxicity were evaluated using CTCAE v3.0. No ≥ grade 4 toxicity was recorded. The most frequent complication after LTA+RT was pneumothorax (G1/G2) in 26/77 (33%) patients [15,16,17], with 16 patients (20.7%) requiring intervention with chest tube placement [16,17]. The second most frequent toxicity was acute respiratory distress (grade not specified) in two patients (2.5%), requiring admission to a respiratory intensive care unit [16]. No ≥ grade 2 acute radiation pneumonitis was recorded. Other complications are reported in Table 1.

### 3.2. Second Group: Radiotherapy Followed by Image-Guided LAT

Thirty-eight patients were evaluated with a male/female ratio of 20/12 (we do not have data on gender in the works of Brooks et al.) [20,21,22]. Median age was described only in two papers and ranged from 64 to 78 years (median 70 years) [20,21]. Initial clinical stage data were reported only in one experience [20], reporting stage I in 5 patients, stage II in 6 patients, and stage III in 1 patient. More details are described in Table 1.

The three papers analyzed a total of 43 LTA sessions after previous radiotherapy in 38 patients. Thirty-one patients underwent EBRT (without any indication on the type of radiotherapy technique) [20,21] and seven SBRT [20,22], but the precise time interval between RT and LAT was not specified except for Brooks et al. [22], where the described median time was 14.9 months.

Only two studies described the type of LAT procedure [19,20]. Twenty-one treatments were RFA procedures [20,21], ten were MWA procedures [20,21], and two were CA procedures [20,21]. Technical parameters were reported only by Leung et al. [21]. The power (W) was 145.5 (range 90–198), the baseline impedance was 59 ohms (range 36–117), the time per lesion was 5 min (range 1–20), and the maximum temperature was 78 °C (range 63–98) for the RFA procedure. Power (W): 52.5 (range 45–60) and time per lesion: 10 min (range 5–10) for the MWA procedure, while the minimum temperature is 128 °C (range −117 to −132) and time per lesion: 8.5 min (range 7–10) for the cryoablation procedure.

Only two papers reported data on RT [20,21]. Variable radiation fractionations were used with a delivered median dose ranging from 50 to 63 Gy. Further data on lesion size and radiological response evaluation are shown in Table 1.

The median follow-up period varied from 8.3 to 69.3 months with an LC rate ranging from 50% to 100% [20,21,22]. Fourteen patients (36.8%) reported local failure after salvage LTA, and in 10 patients (26.3%), a second LTA was required. Among these 10 patients, one recurrence was registered [20,21]. Leung et al. reported a tumor time local progression (TTLP) of 3.3 months (range 1.1–12.2 months), and they showed that a size inferior to 30 mm had a longer TTLP compared to ones with bigger dimensions (23 months vs. 14 months) [20].

In terms of OS, the median OS ranges from 35 to 51.6 months. Cheng et al. [20] reported that a slightly higher mean survival in smaller tumors (<30 mm) could be observed (38 months vs. 35 months). Leung et al. showed rates of CSS at 12, 24, and 60 months of 100%, 56%, and 28%, respectively [21].

The most frequent adverse event after the procedure of LTA was pneumothorax (G1/G2), which was experienced in 8/38 (21%) patients [20,21]. Of these patients, three (20.7%) developed a pneumothorax requiring intervention with chest tube placement [19,20]. Moreover, one patient (2.6%) developed a pseudoaneurysm of a segmental pulmonary artery requiring an embolization intervention (grade 3) [21], and one patient (2.6%) developed a grade 2 pleural effusion that required a thoracentesis [21]. Regarding the RT toxicity profile, there was no acute radiation pneumonitis ≥ grade 2. Other complications are reported in Table 1.

### 3.3. Local Control and Tumor Dimensions

Data on tumor size and LC were both available only in three papers [16,20,21].

Figure 2 depicts the association between lesions size and LC after the combined therapy, using a random-effects model. Lesions up to 30 mm in diameter seem to have a higher possibility to reach local control after the combined therapy, but this was not statistically significant (OR 0.33, CI: 0.06–1.85, *p*: 0.21).

## 4. Discussion

In the last few years, the possibility of a combination strategy between RT and other loco-regional approaches gained more and more attention. In particular, some authors theorized that a possible synergic result combining RT and LAT could be obtained using their different action mechanisms [15,16,17,20,21,22].

Encouraging data were provided by the experiences reported on hepatocellular carcinoma (HCC) or renal cancer. A recent meta-analysis [23] about HCC reported that the combination of SBRT and transcatheter arterial chemoembolization (TACE) might be an excellent choice for HCC with portal vein tumor thrombus (PVTT) rather than SBRT or TACE alone (monotherapy) with significant results in terms of OS and time to progression (TTP). In another setting, Blitzer et al. [24] performed the combination between SBRT and MWA in the treatment of renal cell carcinoma (RCC). The results were promising, indicating that SBRT combined with MW ablation appears to be a safe and feasible therapeutic modality for patients with large volume or vascular invasive RCC with an excellent rate of LC (100%).

These combinations could also be applied for pulmonary lesions due to their characteristics. RT depends on oxygen for cytotoxicity induction and is most effective against well-oxygenated cells, but it is less effective at destroying the hypoxic cells that make up the irregularly vascularized core of a solid neoplasm. Moreover, it is thought that hypoxic cells in the center of many tumors become progressively radiation resistant, contributing to tumor repopulation during RT of extended duration [12,13,14,25,26,27].

In contrast, LTA is most effective at the tumor central zone where the active zone of heating is focused, but it is less effective at damaging the tumor periphery, which tends to have impaired conduction due to the heat sink effect of large, high flow vessels and the insulation effect of aerated lung parenchyma [12,13,14,25,26,27]. Moreover, according to the works of Singh et al. [28,29], the heterogenous temperature distribution in the peripheral regions could also depend on the slight variations in the thermal-diffusion-mediated heat transfer, the blood-perfusion-mediated heat loss across the tumor tissue for the heat sink, and the irregular shape of the lesion.

To our knowledge, this is the first systematic review focused on combined treatment between RT and LTA in lung cancer lesions.

### 4.1. LAT Technique

The three most image-guided lung ablation techniques widely used are RFA, MWA and CA.

The lung is highly susceptible to the RFA technique because the air acts as an insulator, like a low electrical conductivity area. Therefore, it obtains a greater tissue volume ablation for the same energy than any other tissue [30]. The first published retrospective study reported that the 1-, 2-, and 3-year overall survival (OS) rates after the RFA of early NSCLC were 78%, 57% and 36%, respectively, and the local recurrence rates were 12%, 18%, and 21%, respectively [31,32]. According to the prospective multicenter clinical trial (RAPTUR study), NSCLC patients treated with RFA had a 1-year OS of 70% and a 2-year OS of 48% with stage I NSCLC patients having a 2-year OS and cancer-specific survival rate of 75% and 92%, respectively [33]. The main advantage of RFA is the extensive literature, as numerous studies have been conducted to evaluate the safety and efficacy of this treatment [34]. RFA provides an ablation volume with only one probe that can be activated at a time. The RFA is not generally recommended for central or near large vessel tumors or hilar lesions for the heat dissipation effects of neighboring blood vessels. Another disadvantage is that RFA may interfere with the heart’s conduction system and is classically related to cardiac pacemakers’ interference [35]. RFA treatment could be useful in an ideal patient with a peripherical lesion smaller than 3 cm.

Although not as extensively researched as RFA, MWA is becoming increasingly popular for image-guided percutaneous lung ablation. According to the literature, Yang et al. reported a median OS of 33.8 months after MWA among 47 patients with stage I NSCLC. The OS rates at 1, 3, and 5 years were 89%, 43%, and 16%, respectively, and the local control rates at 1, 3, and 5 years were 96%, 64%, and 48%, respectively [36].

Yao et al. found that MVA has similar outcomes to lobectomy for stage I NSCLC, with 1-, 3-, and 5-year OS rates of 100%, 92.6, and 50% for MWA and 100%, 90.7%, and 46.3% for lobectomy, respectively [37]. However, there is evidence that MWA is a promising therapeutic option for advanced lung cancer [38].

MWA may allow the treatment of larger tumors than RFA since tissue impedance does not limit the action of MWA [39]. In particular, MWA may be more effective for central or near large vessel tumors or hilar lesions, as the heat dissipation effect does not interfere with its therapeutic effect. However, it is difficult to control the ablation zone, and there is an increased risk of bronchial fistula when used near the pulmonary hilum. Microwave ablation could be useful in an ideal patient with a peripheral or central lesion larger than 3 cm without limitation regarding pacemaker disposal.

The CA is effective without damaging structures containing a collagenous matrix, such as blood vessels and bronchial tubes, with an advantage for the treatment of tumors near the pulmonary hilum or major vessels treatments [40].

The CA often requires the placement of two or more probes within the lesion, which increases the procedure’s difficulty but allows the customization of the treated area’s morphology. Both MWA and cryoablation (CA) allow for the simultaneous delivery of energy through several probes activated at the same time with a synergistic effect versus subsequent activation of the same probe [35].

However, unlike RFA and MWA, experience with CA is limited. Yamauchi et al. reported the first results of CA for inoperable stage I NSCLC patients with a total of 25 treatments in 22 patients. They found a local control rate of 97%, a median OS of 68 months, and a 3-year OS of 88% [41]. McDevitt et al. reported 1- and 3-year OS rates of 100% and 63%, respectively, in 25 patients with stage I NSCLC treated with CA [42]. One limitation of CA is that the procedure is longer than RFA and MWA with available protocols describing the need for up to three freeze–thaw cycles to achieve a correct ablative treatment [43]. Another disadvantage is that it is not recommended in a patient with coagulopathy due to the increased frequency and severity of pulmonary bleeding and hemoptysis. Cryoablation is an effective alternative in tumors near the great vessels, airways, pericardium, and subpleural lesions, as it tends to cause less pain than RFA and MWA. Another advantage is evaluating the ablation site during the procedure, optimizing the treatment in real time.

According to the literature, these ablative techniques have similar therapeutical results. Therefore, the choice is based on the tumor features and the patient’s characteristics.

Another interesting possible approach is represented by the use of magnetic nanoparticle-based hyperthermia: a new cancer treatment technology that destroys tumors under an external alternating magnetic field [44]. Magnetic nanoparticle-based hyperthermia is a promising therapeutic strategy for non-invasive local tumor treatment, but the clinical use of this remains rare [44,45]. Only one paper [46] resulted from the review of Farzanegan et al. [44] on applying MNPs-based hyperthermia for lung cancer treatment. This study reported that hyperthermia using targeted superparamagnetic iron oxide (SPIO) nanoparticles significantly inhibited in vivo tumor growth. It highlights the potential for developing magnetic hyperthermia as an effective anticancer treatment modality for non-small cell lung cancer treatments [46]. But further studies are needed to evaluate the effectiveness, challenges, and probable defects of magnetic nanoparticle-based hyperthermia for cancer treatment in clinical practice.

### 4.2. RT Techniques

Historically, radiotherapy was delivered with conventional fractionation 1.8–2 Gy for a total dose of 90 Gy. Local recurrence rates were 40%, and 3-year overall and cancer-specific survival rates were 34% and 39%, respectively, which were significantly worse than surgical outcomes [47]. Over the years, SBRT has become the standard treatment in this patient setting, allowing notable improvements to be achieved compared to conventional radiotherapy. SBRT is a non-invasive radiotherapy technique that allows a high biological dose to be administered in a few sessions with extreme precision to a target of limited size thanks to the control of organ movement and an accurate definition of the target volumes. Specifically, SBRT is characterized by the delivery of high doses, greater than 5 Gy per fraction, in a limited number of fractions, and by the rapid drop in dose around the target, resulting in a maximum sparing of surrounding healthy tissues at risk of toxicity. SBRT is a local ablative treatment like a surgical intervention associated with a minimal incidence of local toxicity potentially capable of improving long-term survival without negatively impacting the patient’s quality of life [48,49].

With outstanding outcomes in terms of local control and survival, SBRT is the radiation treatment now used for inoperable primary lung malignancies. Its efficacy has clearly exceeded that of conventional radiotherapy. In the randomized phase III CHISEL study, the risk of disease progression was found to be lower with SBRT (54 Gy in 3 fractions of 18 Gy, or 48 Gy in 4 fractions of 12 Gy) compared to conventional radiotherapy (66 Gy in 33 fractions of 2 Gy) with a favorable toxicity profile (14% vs. 31% HR 0.32 [95% CI 0.13–0.77], *p* = 0.008). There were no treatment-related deaths with only one case of G4 toxicity (dyspnea) in the SBRT arm; grade 3 toxicity was recorded in seven patients (10%) in the experimental arm and in two patients (6%) in the conventional RT arm. Local control at 2 years was 89% in the group of patients undergoing SBRT versus 65% in the patients enrolled in conventional RT [50].

Multiple studies investigated the feasibility and the effectiveness of SBRT for the treatment of lung cancer using a variety of dosing and fractionation schedules.

In the first phase I lung SBRT conducted by Timmerman and colleagues, we reported that doses of 20 Gy per fraction were tolerable and feasible, showing impressive rates of local control [51]. Two and three-year local control rates of 95% and 88%, respectively, were observed in a phase 2 study in which 70 patients were treated with 60–66 Gy [52].

Successively, Timmerman [53] reported in the first multi-institutional phase II trial 3 and 5-year local control rates of 97.6% and 92.7%, respectively, in a cohort of 55 patients treated at a dose of 54 Gy in three fractions with one local failure observed. In another prospective study, Ricardi et al. [10] analyzed 62 patients observing 3.2% of local relapse (2 pts) with a local control rate of 87.8%. In the same papers, the authors showed a significant correlation between tumor diameter and the probability of achieving a complete response, confirming that smaller lesions have a higher chance of being fully controlled and potentially cured.

Furthermore, in a recent review, a direct correlation was demonstrated between the administered dose and local disease control when 100 Gy in BED 10 (Biological Equivalent Dose) was exceeded. From this analysis, it can be seen that the percentage of local relapses is 8% for doses higher than 100 Gy and rises to 27% for lower doses with an impact also on survival (88% vs. 70%) [54].

The prescription dose of SBRT in thoracic tumors is conditioned not only by the tumor volume but also by the site of the disease, as it can influence the response and toxicity of the treatment itself.

### 4.3. Combined Approach

We reported interesting data on LC: the overall LC rate was 74.7% (range: 50–100%) with only 26 pts (24.7%) that reported local failure. These results can be compared with the ones regarding SBRT and LTA alone in the same setting in the current literature (30–55).

In our analysis, the combined treatment shows a limited risk of severe complications. Regarding toxicity profile, we registered pneumothorax (29.8%, with 18 patients requiring interventional therapy), pneumonitis (9.5%), pleural effusion (0.8%), and hemorrhage (0.8%).

Regarding the pneumothorax, the results are in line with the ones described in the literature on LAT alone, ranging from 29% to 34.3%, and about 11% to 12.3% of patients require interventional therapy (chest tube placement) [55]. Pleural effusion generally occurs in 5.2% to 9.6% of patients, and only 0.3% to 0.6% of patients had several pleural effusion requiring intervention in previous studies [56,57].

RT alone can lead to pulmonary toxicity, and the most common side effect of radiation alone is pneumonitis, which has been reported to occur in 5% to 15% of patients. In particular, SBRT presents negligible toxicity: the ratio of patients with grade 3 acute or late adverse event is less than 10%; in our series, no grade 3 or late toxicity was recorded [31,57,58,59].

Tumor size may be still considered a significant factor in the treatment response to combined treatment: we observed that tumors <30 mm had a longer tumor time local progression (TTLP) compared to tumors >30 mm having a shorter TTLP, but it was not statistically significative, which remains a key factor of technical success and clinical efficacy [16,19,20]. In Figure 2, forest plots visually demonstrate the overall relationship between lesions size and LC after the combined therapy. The trend association of tumor size with LC and TTLP obtained in our analysis was concordant with data literature. We have observed for primary NSCLC an average local recurrence of 45.6 ± 4.1 months for lesions <30 mm versus 34 ± 7.8 months for lesions >30 mm [16]. For recurrent NSCLC, we reported a TTLP of 23 months for tumor size ≤30 mm, whereas for tumors >30 mm, it was 14 months [20]. Simon et al. [31] reported a median TTLP for tumors ≤30 mm of 45 months versus 12 months for tumors >30 mm. In another study, Lanuti et al. [60] reported a recurrence rate of 50% for lesions >30 mm compared to 44% for lesions 20–30 mm. Schoellnast et al. [61] observed a median TTLP of 14 months for tumors with a mean size of 28 mm.

It should be noted that we examined case studies including patients who were not deemed the best candidates for surgery, which was most likely due to substantial morbidities, and with a median age ranging from 55 to 93 years.

As describe in Appendix A, the quality of the selected works ranged from the medium to low level. Moreover, we have to acknowledge that concerning the used RT techniques, not all the experiences used SBRT, with many using conventional radiotherapy instead.

Another aspect of these studies that needs further investigation is the sequence in which LAT and RT should be combined. In all the reported experiences, LAT was followed by RT; however, the hypoxia provided by LAT could make the cancerous tissue more radioresistant. Thus, RT, and in particular SBRT, should be performed before the LAT for radiobiological reasons.

Another point we have to consider in evaluating these results is the effect that blood perfusion can have on the efficacy of thermal ablation cancer treatments due to the heat-sink effect. This is due to heterogeneously perfused tumor regions that cause such a variability in thermal response to heating and thermal ablation, playing a crucial role in heat transfer within tissues. In fact, a heterogeneous blood perfusion can lead to significant variations in temperature distribution within tumors, and regions with lower blood perfusion may exhibit different sensitivity to therapies compared to areas with higher perfusion [62].

Even though the present systemic review had some limitations (small sample sizes, retrospective nature of the considered studies, their heterogeneity in terms of radiation treatment schedules and LTA and the short follow-up period), the data showed interesting results in terms of LC and toxicity. Our review could be considered a starting point for a further randomized controlled clinical study regarding the combination between RT and LAT in the treatment of primary or secondary lung cancer. However, we must remember that the key element in this treatment strategy should always be a harmonized multidisciplinary approach.

## 5. Conclusions

The proposed intervention demonstrated encouraging local control rates as well as low toxicity profiles. Despite these promising outcomes, it should be noted that these data come from retrospective studies with a significant level of heterogeneity, making it impossible to recommend an a priori strategy involving RT + LTA for patients in this context. While we await further randomized trials to verify this method, we propose a case-by-case evaluation based on tumor and patient characteristics.

## Figures and Tables

**Figure 1 cancers-15-05869-f001:**
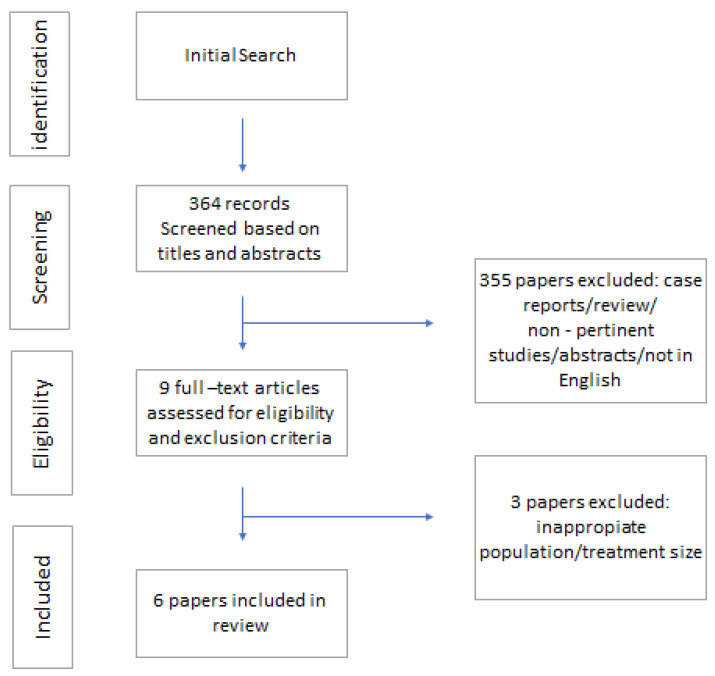
PRISMA literature search.

**Figure 2 cancers-15-05869-f002:**
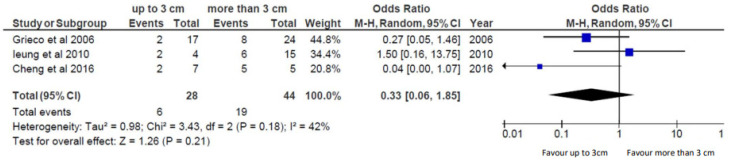
Forest plot investigating the relationship between lesions size and LC after the combined therapy [16,20,21].

**Table 1 cancers-15-05869-t001:** Selected studies characteristics.

Author	Study Design	No of Patients	Sex	Median Age	Stage	Treat.	RT	LTA	Time Start II Treatment	Size	Dose RT	Median Follow-Up	PFS	IR	LP	LC Rate	OS	TOX
Steber [15]	PII	12	8 M/ 4 W	71 (60–93)	IA (11) IB (1)	LAT + RT	EBRT	RFA 12	36 days (27–60)	≤35 mm	HFRT (10 pts) 70.2 Gy-fSBRT (3 pts) 54 Gy	51.7 (12.3–130.9)	37.8 months	PET CT	3 (25)% at 6.8, 49.7 and 135.4 m CILP at 5 year 16.7%	75%	53.6 months (median)	G3 skin (1 pts 8%—full thickness thermal burn) G1 hemorrhage (3 pta 25%) G2 hemorrhage (1 pts 8%) G1 Pnx (4 pts 33%) G2 pneumothorax (4 pts 33%) G1 pneumonitis (10 pts 83%) G2 gastrointestinal (1 pts 8%)
Grieco [16]	R	41	24 M/17 W	76 (55–81)	IA (21) IB (17) II B (3)	LAT + RT (27 PTS) LAT + BT (14 pts)	EBRT (27)	RFA ablation: 37 MWA ablation: 4	EBRT 24 days (5–53) after LTA	≤30 mm (17 pts) ≥30 mm (24 pts)	CFRT 66 Gy (26 pts)—CFRT 50 Gy 1 pts IRT-BT: 13 pts HDR ^192^Ir/1 pts LDR ^125^I	19.5 (1.0–73)	NA	PET CT 41/41	10/41 (24.4%) 2/17 < 30 mm 8/24 > 30 mm; average local recurrence: (11.8%) 45.6 ± 4.1 m. <30 mm vs. (33%) 34 ± 7.8 > 30 mm	75.6%	Average OS 34.7 ± 5.4 m for LTA + BT (14 pts) vs. 42 ± 6 m for LTA + RT (27 pts)	Pneumothorax 15 pts (G2 9/15 22%—chest tube placement)—Acute respiratory distress 2 pts/41 (4.9%) (admission to respiratory intensive care unit)
							IRT-BT (14)		IRT-BT 1–2 h after RFA								CSS rates 97.6% at 6 m, 86.8% at 12 m and 57.1% at 36 m—average survival time 44.4 ± 5.3 months < 30 mm vs 34.6 ± 7 > 30 mm	
Dupuy [17]	R	24	10 M/14 W	76 (58–85)	IA (11) IB (10) NA (3)	LAT + RT	EBRT	RFA 24	NA	median size: 34 mm (1.5–7.5)	CFRT 66 Gy	26.7 (6–65)	NA	PET CT (17 pts) CT (7 pts)	2/24 pts (8.3%)	91.7%	Mean follow-up period 26.7 CSS at 12, 24, 60 m was 83%, 50%, 39%; according to tumor stage CSS: stage IA: 12, 24, and 56 m were 92%, 62%, and 46% stage IB: 12, 24, 60 months were 73%, 42%, and 31%, respectively.	Pneumothorax 7 pts-29% (G2 3 pts—12.5%—chest tube placement), Radiation fibrosis 2 pts (8.3%)
Cheng [20]	R	12 pts	8 M/F 4	71 ± 7	I (5) II (6) III (1)	LTA local salvage	RT (50–63 Gy) 11 EBRT + 1 fSBRT	RFA 4/MWA 13	NA	34 mm ± 13 mm	RT (50–63 Gy)	19 ± 11 months	NA	CT within 1 month and 3 months and PET TC every 6 m.	TTLP 14 median months: 6/12 pts --> 5 re-ablation → 1/5 pts III ablations (tumor size ≤ 30 mm TTLP 23 months—tumor size ≥ 30 mm TTLP 14 mm), local progression rate at 1 year was 45%	50%	35 (median CI 12.58) mean survival tumor < 30 mm: 38 m, tumor > 30 mm: 35 m	Pneumothorax 5 post- RFA 29% (2 chest tube placement (12%))
Leung [21]	R	20	12 M/8 F	70.5	IA-IV	LTA local salvage	RT 60.4	RFA 17/MWA 6/CA 2	NA	40 mm	RT 60.4 Gy (50.4–77.4)	3.1 to 67.7 m (median, 10.4 m)	NA	CT within 1 month and 3 months and PET TC every 6 m.	8/20 (40%) TTLP 3.3 months (1.1–12.2) 5/8 re-ablation	60%	13.1 ± SE1.2 m 10.4 months (3.1–67.7) CSS at 6.12 24 months was 100%, 56%, 28%	G3 pseudoaneurysm/hemoptysis 1 pts (4%) (embolization)—G2 pneumothorax 1 pts (4%) (chest tube placement)—G2 pleural effusion 1 pts (4%) (chest tube placement—G2 empyema 1 pts [4%]
Brooks [22]	R	6	NA	NA	NA	LTA local salvage	SABR	NA	14.9 (1.5–91.9)	NA	NA	38.5 (19.9–69.3)	NA	NA	0/6	100%	51.6 m	NA

Legend: R (Retrospective), PII (Phase II), M (Male), W (Woman), NA (Not available), Treat. (Treatment), IR (Imaging response), EBRT (External beam radiotherapy), IRT-BT (Interventional radiotherapy–Brachytherapy), LTA (Local thermal ablation), RFA (Radiofrequency ablation), MWA (Microwave ablation) CA (Cryoablation), HFRT (Hypofractionated radiotherapy), CFRT (Conventional fractionated radiotherapy), fSBRT (Fractionated Stereotactic Body Radiotherapy), LP (Local progression), LC (Local control), CSS (Cancer-specific survival), OS (Overall survival), TOX (Toxicity), TTLP (Time To Local Progression), CT (Computed Tomography), PET (Positron Emission Tomography). SABR: Stereotactic ablative radiotherapy.

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
