# Peer review of "Combination of Local Ablative Techniques with Radiotherapy for Primary and Recurrent Lung Cancer: A Systematic Review"

_cancers, 2023, doi:10.3390/cancers15245869_

Round 1
Reviewer 1 Report
Comments and Suggestions for Authors
I read with great notice the interesting manuscript entitled: "Combination of local ablative techniques (LAT) with Radiotherapy (RT) for primary and recurrent lung cancer: A Systematic Review"
It is a well written paper, with analysis according to PRISMA guidelines.
I have the impression that the manuscript diserves publication.
I suggest for the authors to take into account the following:
The sequence of SBRT and LAT should discussed. The ypoxia provied by the LAT is making the cancerous tissue really radioresistant. Thus the procedure should be first SBRT and (in case of not complete responce) thereafter LAT for radiobiological reasons.
Comments on the Quality of English Language
Minor editing
Author Response
Q1: I read with great notice the interesting manuscript entitled: "Combination of local ablative techniques (LAT) with Radiotherapy (RT) for primary and recurrent lung cancer: A Systematic Review". It is a well written paper, with analysis according to PRISMA guidelines. I have the impression that the manuscript diserves publication. I suggest for the authors to take into account the following: The sequence of SBRT and LAT should discussed. The hypoxia provied by the LAT is making the cancerous tissue really radioresistant. Thus the procedure should be first SBRT and (in case of not complete responce) thereafter LAT for radiobiological reasons.
A1: we thank the reviewer for his comments. Following his insightfull suggestion, we took into account the sequence issue in the discussion section (page 15, lines 492-496): “Another aspect of these studies that need a further investigation is the sequence in which LAT and RT should be combined. In all the reported experiences, LAT was followed by RT; however, the hypoxia provided by LAT could make the cancerous tissue more radioresistant. Thus, RT, and in particular SBRT, should be performed before the LAT for radiobiological reasons.”
Reviewer 2 Report
Comments and Suggestions for Authors
-
One of the primary concerns with this review paper is the lack of clarity in defining the scope and objectives of the study. The authors have not clearly outlined the specific research questions or goals they aimed to address. As a result, the paper appears to be a collection of loosely connected ideas rather than a cohesive review.
-
Another major issue is the lack of rigorous methodology. The paper does not provide any information regarding the systematic search and selection of relevant articles or studies. A comprehensive review should demonstrate a well-defined methodology to ensure the inclusion of relevant and reliable sources.
-
The review paper also suffers from a lack of critical analysis and evaluation of the reviewed technologies. Merely describing and summarizing the existing technologies without offering any meaningful insights or comparisons diminishes the value of the review. The paper should have critically examined the strengths, limitations, and potential areas of improvement for each technology.
-
The overall organization and structure of the paper are inadequate. The flow of ideas is unclear, and there is a lack of coherence between sections. The paper should have presented a clear introduction, outlined the main themes or categories of technologies, and provided a concise summary or conclusion to tie the information together. The following papers are good examples:
https://doi.org/10.1155/2022/5052435
https://doi.org/10.3390/bioengineering10040495
-
Author Response
Q1: One of the primary concerns with this review paper is the lack of clarity in defining the scope and objectives of the study. The authors have not clearly outlined the specific research questions or goals they aimed to address. As a result, the paper appears to be a collection of loosely connected ideas rather than a cohesive review.
A1: We thank the reviewer for the comment. As described in the introduction, we observed only few and heterogenous experiences regarding the combination of RT techniques with LAT in the treatment of primary and recurrent lung cancer. Because of this, we tried to collect with this paper all the experiences on this setting, focusing on the local control rate and toxicity profile. However, as suggested, we revised the introduction and the methods sections in order to be clearer on the review goal.
Q2: Another major issue is the lack of rigorous methodology. The paper does not provide any information regarding the systematic search and selection of relevant articles or studies. A comprehensive review should demonstrate a well-defined methodology to ensure the inclusion of relevant and reliable sources.
A2: We thank the reviewer for the suggestion. As requested we restructured the methods section, adding more data, especially the quality and bias assessment.
Q3: The review paper also suffers from a lack of critical analysis and evaluation of the reviewed technologies. Merely describing and summarizing the existing technologies without offering any meaningful insights or comparisons diminishes the value of the review. The paper should have critically examined the strengths, limitations, and potential areas of improvement for each technology.
A3: We thank the reviewer for the insightful comment. Following the suggestion, we revised the discussion section, trying to be more critical in our analysis of the different treatment strategy in the discussion section.
Q4:The overall organization and structure of the paper are inadequate. The flow of ideas is unclear, and there is a lack of coherence between sections. The paper should have presented a clear introduction, outlined the main themes or categories of technologies, and provided a concise summary or conclusion to tie the information together. The following papers are good examples.
A4: We thanks the reviewer for the analysis. As suggested (and already overmentioned), we revised extensively revised the manuscript both in terms of contents and grammar.
Reviewer 3 Report
Comments and Suggestions for Authors
Peer Review Report
Manuscript ID: Cancers-2663284
Title: Combination of local ablative techniques (LAT) with Radiotherapy (RT) for primary and recurrent lung cancer: A Systematic Review
The study “Combination of local ablative techniques (LAT) with Radiotherapy (RT) for primary and recurrent lung cancer: A Systematic Review” by Paolo et al. 2023 lies within the Journal scope of Cancers. The study summarizes comprehensive review of six studies that accounted a total of 115 patients and 119 lesions in the treatment of primary and recurrent lung cancer.
1. The title should not have abbreviations and avoid usage such as LAT and RT.
2. The authors should elaborate abbreviations such as LTA-RT, NSCLC, SBRT, F-UP, TACE, RCC etc. Avoid usage of such abbreviations in Abstract.
3. Lines 65-69
Indeed, Radiotherapy (RT) and LTA rely on totally different mechanisms: the first one is most effective against well-oxygenated cells in the peripherical areas of the tumor and less effective at eradicating more hypoxic cells in centrally located tumor, whereas LAT targets the core but is less effective in the peripheric areas of the tumor due to increasing heat sink effects [12, 13, 14]. In reference to lines 65-69, it is advised to include recent developments on stated claims: 1. well oxygenated cells in the peripheral areas of tumor……..[Incorporating vascular-stasis based blood perfusion to evaluate the thermal signatures of cell-death using modified Arrhenius equation with regeneration of living tissues during nanoparticle-assisted thermal therapy], 2. Less effective at eradicating more hypoxic cells in centrally located tumor and local thermal ablation is less effective in peripheral tumor regions due to heat-sink effects. This is due to heterogeneously perfused tumor regions that causes such variability in response to heating and thermal ablation [Modified Pennes bioheat equation with heterogeneous blood perfusion: A newer perspective]. This affects thermal ablation beyond tumor rims [Pre-operative Assessment of Ablation Margins for Variable Blood Perfusion Metrics in a Magnetic Resonance Imaging Based Complex Breast Tumour Anatomy: Simulation Paradigms in Thermal Therapies].
4. The Manuscript have a poor introduction, lacks motivation and rationale to perform this comprehensive literature review. The authors can use suggestion given by Comment #3 to improve Introduction section. There are lot of shortcomings of thermal ablation therapies and synergistic treatments are needed.
5. Lines 57-58:One of them is the percutaneous image-guided local tumor ablation (LTA), including radiofrequency (RFA), microwave ablation (MWA) or cryoablation (CA). The study should include magnetic nanoparticles induced thermal ablation (magnetic field based therapy) under image guidance [Biological heat and mass transport mechanisms behind nanoparticles migration revealed under microCT image guidance].
6. Lines 112: Language correction is advised for unstructured sentences. The entire manuscript must be proof-read to avoid any technical jargons and long unstructured sentences. patient’s exitus survival
7. Lines 131-132: What is the difference between retrospective and prospective?
8. Restructure Table 1 as it is difficult to read and follow.
9. F-UP: Is this refers to Follow Up?
10. Figure 2 has poor resolution. Improve resolution of the figure and update.
11. What statistical tests were used to analyze the data? Why those tests were used to complete the analysis?
12. Discussion section should elaborate the studies reported in Table 1 and comparison should be made in context to the reported therapies.
13. Line 266-269
In contrast, LTA is most effective at the tumor central zone where the active zone of heating is focused, but it is less effective at damaging the tumor periphery, which tends to have impaired conduction due to the heat sink effect of large, high flow vessels and the insulation effect of aerated lung parenchyma. Elaborate this statement in context of study by Zhu and co-workers where influence of blood perfusion and interstitial space of tumor is quantitatively explained [Quantitative evaluation of effects of coupled temperature elevation, thermal damage, and enlarged porosity on nanoparticle migration in tumors during magnetic nanoparticle hyperthermia]
14. Line 317: Candidable is not any word. Proof-read your writing.
15. We advise authors to revise Conclusions section and qualitatively explain in reference to quantitative inferences.
We are looking forward to reviewing revised version of the Manuscript.
Comments on the Quality of English LanguageAvoid technical jargons and write scientific details in a structured way so that one can understand the key message from this comprehensive review.
Author Response
Q1. The title should not have abbreviations and avoid usage such as LAT and RT.
A1: We thank the reviewer for the suggestion. We modified the title as requested.
Q2: The authors should elaborate abbreviations such as LTA-RT, NSCLC, SBRT, F-UP, TACE, RCC etc. Avoid usage of such abbreviations in Abstract.
A2: We thank the reviewer for the suggestion. We modified the text as suggested
Q3. Lines 65-69 Indeed, Radiotherapy (RT) and LTA rely on totally different mechanisms: the first one is most effective against well-oxygenated cells in the peripherical areas of the tumor and less effective at eradicating more hypoxic cells in centrally located tumor, whereas LAT targets the core but is less effective in the peripheric areas of the tumor due to increasing heat sink effects [12, 13, 14]. In reference to lines 65-69, it is advised to include recent developments on stated claims: 1. well oxygenated cells in the peripheral areas of tumor……..[Incorporating vascular-stasis based blood perfusion to evaluate the thermal signatures of cell-death using modified Arrhenius equation with regeneration of living tissues during nanoparticle-assisted thermal therapy], 2. Less effective at eradicating more hypoxic cells in centrally located tumor and local thermal ablation is less effective in peripheral tumor regions due to heat-sink effects. This is due to heterogeneously perfused tumor regions that causes such variability in response to heating and thermal ablation [Modified Pennes bioheat equation with heterogeneous blood perfusion: A newer perspective]. This affects thermal ablation beyond tumor rims [Pre-operative Assessment of Ablation Margins for Variable Blood Perfusion Metrics in a Magnetic Resonance Imaging Based Complex Breast Tumour Anatomy: Simulation Paradigms in Thermal Therapies].
A3: We thank the reviewer for the insightful suggestion. We added this point and the related references in the discussion section (page 11, lines 317-323)
Q4: The Manuscript have a poor introduction, lacks motivation and rationale to perform this comprehensive literature review. The authors can use suggestion given by Comment #3 to improve Introduction section. There are lot of shortcomings of thermal ablation therapies and synergistic treatments are needed.
A4: Thanks for the interesting suggestion. As required we extensively revised the paper
Q5: Lines 57-58:One of them is the percutaneous image-guided local tumor ablation (LTA), including radiofrequency (RFA), microwave ablation (MWA) or cryoablation (CA). The study should include magnetic nanoparticles induced thermal ablation (magnetic field based therapy) under image guidance [Biological heat and mass transport mechanisms behind nanoparticles migration revealed under microCT image guidance].
A5: We thank the reviewer for the interesting suggestion. As described in the results section, none of the reported experiences showed an association between magnetic nanoparticle and RT. However, we added cited this strategy in the discussion section (page 13, lines 387-398): “Another interesting possible approach is represented by the use of magnetic nano-particle-based hyperthermia: a new cancer treatment technology that destroys tumors under an external alternating magnetic field [44]. Magnetic nanoparticle-based hyper-thermia is a promising therapeutic strategy for non-invasive local tumor treatment, but the clinical use of this remains rare [45,46]. Only one paper (Sadhukha et al., 2013) resulted from the review of Farzanegan et al., 2023 on applying MNPs-based hyperthermia for lung cancer treatment. This study reported that hyperthermia using targeted superpara-magnetic iron oxide (SPIO) nanoparticles significantly inhibited in vivo tumor growth. It highlights the potential for developing magnetic hyperthermia as an effective anticancer treatment modality for non-small cell lung cancer treatments. [47]. But further studies are needed to evaluate effectiveness, challenges, and probable defects of magnetic nanoparti-cle-based hyperthermia for cancer treatment in clinical practice..”
Q6: Lines 112: Language correction is advised for unstructured sentences. The entire manuscript must be proof-read to avoid any technical jargons and long unstructured sentences. patient’s exitus survival
A6: As requested a language correction was performed
Q7. Lines 131-132: What is the difference between retrospective and prospective?
A7: A retrospective study is performed “a posteriori”, using information on events that have taken place in the past; while a prospective study is performed “a priori”, and evaluate outcomes during the study period and relates this to other factors such as suspected risk or protection factor(s). The papers considered in this review were defined as retrospective or prospective by their own authors.
Q8: Restructure Table 1 as it is difficult to read and follow.
A8: We thank the reviewer for the suggestion. The table in question was created following the Journal requests and guidelines, so we ask to the editor if it could be possible using another format.
Q9. F-UP: Is this refers to Follow Up?
A9: We confirm the author query. We modified the text in order to avoid this abbreviation.
Q10. Figure 2 has poor resolution. Improve resolution of the figure and update.
A10: As requested we improved Figure2 resolution.
Q11. What statistical tests were used to analyze the data? Why those tests were used to complete the analysis?
A11: Thanks for the on-spot comment. A forest plot for a post-hoc meta-analysis to display the association between lesions size and LC after the combined therapy was generated. We used random-effects models because there was great subjectivity given the lack of related control groups in the non-comparative studies, and a tendency towards high heterogeneity we added the requested data in the statistical analysis sub-section of the methods section (page 3, lines 126-130): “A forest plot for a post-hoc meta-analysis to display the association between lesions size and LC after the combined therapy was generated. We used random-effects models because there was great subjectivity given the lack of related control groups in the non-comparative studies, and a tendency towards high heterogeneity.”
Q12. Discussion section should elaborate the studies reported in Table 1 and comparison should be made in context to the reported therapies
A12: We thank the reviewer for the comment. As requested, we extensively revised the discussion section.
Q13. Line 266-269 In contrast, LTA is most effective at the tumor central zone where the active zone of heating is focused, but it is less effective at damaging the tumor periphery, which tends to have impaired conduction due to the heat sink effect of large, high flow vessels and the insulation effect of aerated lung parenchyma. Elaborate this statement in context of study by Zhu and co-workers where influence of blood perfusion and interstitial space of tumor is quantitatively explained [Quantitative evaluation of effects of coupled temperature elevation, thermal damage, and enlarged porosity on nanoparticle migration in tumors during magnetic nanoparticle hyperthermia]
A13: We thank the reviewer for the suggestion. We elaborate the concept as required (page 11, lines 314-321): “In contrast, LTA is most effective at the tumor central zone where the active zone of heating is focused, but it is less effective at damaging the tumor periphery, which tends to have impaired conduction due to the heat sink effect of large, high flow vessels and the insulation effect of aerated lung parenchyma. [12, 13, 14, 25, 26, 27]. Moreover, according to the works of Singh et al [28,29], the heterogenous temperature distribution in the pe-ripheral regions could also depend on the slight variations in the thermal-diffusion mediated heat-transfer, the blood-perfusion-mediated heat loss across the tumor tissue for the heat sink, and the irregular shape of the lesion.”
Q14. Line 317: Candidable is not any word. Proof-read your writing.
A14: thanks for the suggestion. We modified the paragraph and adjusted the sentence (page 14, lines 485-487): “It should be noted that we examined case studies including patients who were not deemed the best candidates for surgery, most likely due to substantial morbidities, and with a median age ranging from 55 to 93 years.”
Q15. We advise authors to revise Conclusions section and qualitatively explain in reference to quantitative inferences.
A15: thanks for the insightful comment. We revised the conclusions section as requested (page 17, lines 558-563): “The proposed intervention demonstrated encouraging local control rates as well as low toxicity profiles. Despite these promising outcomes, it should be noted that these data come from retrospective studies with a significant level of heterogeneity, making it impossible to recommend an a priori strategy involving RT+LTA for patients in this context. While we await further randomized trials to verify this method, we propose a case-by-case evaluation based on tumor and patient characteristics.”
Round 2
Reviewer 2 Report
Comments and Suggestions for Authors
From the response letter, the paper has been well revised, and the current version of the manuscript is acceptable for publication.
Author Response
Tanks for your comments
Reviewer 3 Report
Comments and Suggestions for Authors
Peer Review Report
Manuscript ID: Cancers-2663284
Title: Combination of local ablative techniques with Radiotherapy for primary and recurrent lung cancer: A Systematic Review
The study “Combination of local ablative techniques with Radiotherapy for primary and recurrent lung cancer: A Systematic Review” by Paolo et al. 2023 have been substantially improved following the first peer-review process. The authors have addressed most of the comments. However, the manuscript still needs minor revision before it should be recommended for the publication the journal of Cancers. Currently the study lacks discussion on blood perfusion that affects the efficacy of cancer treatments. Elaborate discussion on blood perfusion of tumors that affects thermal ablation due to heat-sink effects in context of recent studies under discussion section. This is due to heterogeneously perfused tumor regions that causes such variability in thermal response to heating and thermal ablation. Blood perfusion plays a crucial role in heat transfer within tissues. Heterogeneous blood perfusion can lead to significant variations in temperature distribution within tumors. Regions with lower blood perfusion may exhibit different sensitivity to therapies compared to areas with higher perfusion. Support the heat-sink effects with this recent development [https://doi.org/10.1016/j.ijheatmasstransfer.2023.124698]. We are looking forward to reviewing revised version of the Manuscript.
Lines 64-66 and 266-269
Less effective at eradicating more hypoxic cells in centrally located tumor and local thermal ablation is less effective in peripheral tumor regions due to heat-sink effects.
In contrast, LTA is most effective at the tumor central zone where the active zone of heating is focused, but it is less effective at damaging the tumor periphery, which tends to have impaired conduction due to the heat sink effect of large, high flow vessels and the insulation effect of aerated lung parenchyma.
Comments on the Quality of English LanguageAvoid technical jargons in your writing.
Author Response
We thanks the reviewer for the comment and for the provided reference. As suggested, we added in the discussion section a paragraph regrading this issue, underling the possible influence of heat-sink effect on the results (page 15, lines 497-503)